# Omega-3 Fatty Acid Supplementation, Pro-Resolving Mediators, and Clinical Outcomes in Maternal-Infant Pairs

**DOI:** 10.3390/nu11010098

**Published:** 2019-01-05

**Authors:** Tara M. Nordgren, Ann Anderson Berry, Matthew Van Ormer, Samuel Zoucha, Elizabeth Elliott, Rebecca Johnson, Elizabeth McGinn, Caleb Cave, Katherine Rilett, Kara Weishaar, Sai Sujana Maddipati, Harriet Appeah, Corrine Hanson

**Affiliations:** 1Division of Biomedical Sciences, School of Medicine, University of California, Riverside, CA 92521, USA; 2Pulmonary, Critical Care, Sleep & Allergy Division, Department of Internal Medicine, University of Nebraska Medical Center, Omaha, NE 68198-5910, USA; 3Department of Pediatrics, College of Medicine, University of Nebraska Medical Center, Omaha, NE 68198-1205, USA; alanders@unmc.edu (A.A.B.); matthew.vanormer@unmc.edu (M.V.O.); samuel_b_zoucha@rush.edu (S.Z.); eelliott@childrensnational.org (E.E.); Rebecca.johnson@unmc.edu (R.J.); elizabeth.mcginn@unmc.edu (E.M.); caleb.cave@unmc.edu (C.C.); katherine.rilett@unmc.edu (K.R.); Kara.weishaar@unmc.edu (K.W.); sujanamaddipati@creighton.edu (S.S.M.); harriet.appeah@unmc.edu (H.A.); 4Medical Nutrition Education Division, College of Allied Health Professions, University of Nebraska Medical Center, Omaha, NE 68198-4045, USA; ckhanson@unmc.edu

**Keywords:** omega-3 fatty acid, docosahexaenoic acid (DHA), specialized pro-resolving lipid mediator (SPM), resolvins, pregnancy, fetal health

## Abstract

Omega (n)-3 fatty acids are vital to neonatal maturation, and recent investigations reveal n-3 fatty acids serve as substrates for the biosynthesis of specialized pro-resolving lipid mediators (SPM) that have anti-inflammatory and immune-stimulating effects. The role SPM play in the protection against negative maternal-fetal health outcomes is unclear, and there are no current biomarkers of n-3 fatty acid sufficiency. We sought to ascertain the relationships between n-3 fatty acid intake, SPM levels, and maternal-fetal health outcomes. We obtained n-3 fatty acid intake information from 136 mothers admitted for delivery using a food frequency questionnaire and measured docosahexaenoic acid (DHA)-derived SPMs resolvin D1 (RvD1) and RvD2 in maternal and cord plasma. We found significantly elevated SPM in maternal versus cord plasma, and increased SPM levels were associated with at-risk outcomes. We also identified that increased DHA intake was associated with elevated maternal plasma RvD1 (*p* = 0.03; R^2^ = 0.18) and RvD2 (*p* = 0.04; R^2^ = 0.20) in the setting of neonatal intensive care unit (NICU) admission. These findings indicate that increased n-3 fatty acid intake may provide increased substrate for the production of SPM during high-risk pregnancy/delivery conditions, and that increased maternal plasma SPM could serve as a biomarker for negative neonatal outcomes.

## 1. Introduction

The health benefits of diets high in omega-3 fatty acids are well established, including in the context of maternal-fetal health [1,2,3,4,5]. However, the mechanisms underlying these benefits are less clear. Polyunsaturated fatty acids have recently been found to serve as substrates for the biosynthesis of specialized pro-resolving lipid mediators (SPM), whereby unesterified fatty acids are enzymatically modified (e.g., oxidation reactions) to yield bioactive lipid metabolites that regulate inflammation resolution [6,7]. SPM are primarily derived from the omega (n)-3 fatty acids docosahexaenoic acid (DHA; producing the resolvin D [RvD]-, maresin-, and protectin- series SPM) and eicosapentaenoic acid (producing the resolvin E [RvE]-series SPM), while the lipoxin series of SPM are derived from n-6 fatty acids [8]. Preclinical investigations reveal these SPM potently limit inflammatory processes and promote tissue restitution while having immune-stimulatory and protective effects against infection [6,7,8,9,10,11,12]. Interestingly, SPM are present in human breast milk in up to 100-fold higher levels than typically found in adult serum during the first month of lactation [13,14,15]. Furthermore, infants transitioned from a soybean oil-based lipid emulsion (SOLE) to a fish oil-based lipid emulsion (FOLE; high in n-3 fatty acids) exhibit increased anti-inflammatory, pro-resolving lipid mediator production [16]. These findings indicate an important biological role for SPM in infant health [17]. However, it is not currently known how SPM contribute to maternal-fetal health, infant health, and disease risk. Furthermore, dietary reference intakes for n-3 fatty acids have not been standardized for pregnant women or women of childbearing age, and US population-level analyses indicate that these women have n-3 fatty acid intakes that fall well below recommendations [18,19]. It is unclear how maternal omega-3 fatty acid intake correlates with SPM production during pregnancy and infant SPM levels, particularly during inflammatory outcomes when SPM would likely be endogenously produced. 

To address these gaps of knowledge, we obtained n-3 fatty acid intake information from pregnant women admitted for delivery to a Midwestern academic medical center using food frequency questionnaires and collected plasma samples from mothers following admission and from cord blood upon delivery. We assessed levels of the DHA-derived SPM RvD1 and RvD2 in these blood samples and compared them to adverse infant health outcomes, including neonatal intensive care unit (NICU) admission and preterm delivery. We also assessed how maternal n-3 supplement usage and DHA intake correlated with circulating SPM levels. Through these investigations, we identified that circulating maternal plasma levels of RvD1 and RvD2 are increased compared to infant cord plasma levels. Furthermore, RvD1 and/or RvD2 levels were increased in maternal-fetal pairs with NICU admission upon delivery and preterm delivery, with increases in these negative outcomes associated with maternal DHA intake. These findings suggest that increased n-3 fatty acid intake may provide increased substrate for SPM production during high-risk pregnancy/delivery conditions and increased maternal plasma SPM could serve as a biomarker for these at-risk conditions.

## 2. Materials and Methods

### 2.1. Cohort Recruitment and Collection of Plasma Samples

This project was conducted at the University of Nebraska Medical Center (UNMC) and its’ clinical partner, Nebraska Medicine. UNMC is the only public academic health science center in the State of Nebraska, and the major teaching and referral center serving rural and urban populations in Nebraska and the Midwest. The IRB for the University of Nebraska Medical Center provided human subjects approval under IRB# 115-12-EP. No power analysis for overall study recruitment was calculated, as this is an ongoing longitudinal cohort implemented with the intent of assessing and monitoring nutritional status in a Midwestern maternal-child cohort. Recruitment included oversampling of NICU admissions for sample size purposes. After informed consent, peripheral blood (ethylene diamine tetraacetic acid plasma (EDTA plasma)) was collected from 136 mothers when admitted for delivery. Cord blood (EDTA plasma) samples were collected from 138 infants at the time of delivery. Exclusion criteria included congenital abnormalities, gastrointestinal (GI), liver, or kidney disease, or inborn errors of metabolism in the infant or the mother. Samples were protected from heat and light immediately upon collection and processed/frozen at −80 degrees. Demographic information was collected on the mother and infant after enrollment. Additional clinical outcomes collected included delivery method, admission to the NICU, a clinical diagnosis of or suspected chorioamnionitis in the mother, a clinical diagnosis of Respiratory Distress Syndrome (RDS) in the infant, and administration of antibiotics to the infant. Prenatal vitamin usage was collected from the medical record, which includes prenatal visits, and mothers were asked to take a Willet Food Frequency Questionnaire (FFQ) during admission for labor and delivery. The FFQs were analyzed by trained personnel at the Harvard School of Public Health, and individualized nutrient intakes were calculated based on the known nutrient content of foods. Subjects with average daily intakes outside a biologically plausible range (less than 700 kcal or greater than 7000 kcal) were excluded from the analyses.

### 2.2. Lipid Mediator Measurement

Maternal and cord plasma levels of RvD1 and RvD2 were measured using commercially available enzyme immunoassays (Cayman Chemical, Ann Arbor, MI, USA) performed according to manufacturer’s directions. Assay results calculations were performed using the Cayman Chemical-provided enzyme immunoassay data analysis computer spreadsheet.

### 2.3. Statistical Analyses

Descriptive statistics were calculated for all variables. To test for normal distribution of the maternal and cord blood plasma levels of RvD1 and RvD2, column statistics were run in GraphPad Prism software (San Diego, CA, USA) including the D’Agostino & Pearson omnibus normality test. Samples did not pass the normality test. Thus, maternal and cord blood plasma levels of RvD1 and RvD2 were compared to pregnancy/delivery conditions using nonparametric two-tailed Mann Whitney tests. Correlations between RvD1 and RvD2 were evaluated using nonparametric Spearman correlation tests. Linear regression models were used to evaluate the association between SPM levels and DHA intakes. Receiver-operating characteristic (ROC) curves were generated to assess the capacity of the maternal plasma SPM levels to discriminate between term/preterm delivery or NICU admission outcomes. Resulting *p* values ≤ 0.05 were considered to be significant.

## 3. Results

### 3.1. Maternal and Infant Cohort Characteristics

The final number of participants included in the study was 136 mothers and 138 infants. Mother and infant characteristics are summarized in Table 1. Within this cohort, the mean DHA intake (including diet and supplementation) was 119 mg/day, with 18% of women reporting the use of n-3 fatty acid-containing supplements. By comparison, the recommended minimum DHA intake suggested by the Workshop on the Essentiality of and Recommended Dietary Intakes for Omega-6 and Omega-3 Fatty Acids is 300 mg/day for pregnant women [20].

### 3.2. SPM Levels in Maternal and Cord Blood Plasma upon Admission for Childbirth

Recent preclinical and clinical investigations indicate a potential increase in circulating lipid mediator precursors in maternal blood in late pregnancy and during childbirth [21,22]. However, the levels of RvD1 and RvD2 in maternal-fetal circulation at the time of delivery have not been reported. Using maternal and cord blood plasma samples taken upon admission for delivery and at childbirth, respectively, we assessed the levels of DHA-derived SPM RvD1 and RvD2. As shown in Figure 1A,B, the median maternal blood plasma levels of RvD1 and RvD2 were 2617 pg/mL and 819.0 pg/mL, respectively. The median cord blood plasma levels of RvD1 and RvD2 were 61.0 pg/mL and 6.4 pg/mL, respectively. Maternal plasma levels of RvD1 and RvD2 were both significantly higher than measured cord blood plasma levels (*p* < 0.0001 for each analysis). As shown in Figure 1C, maternal RvD1 and RvD2 levels were strongly correlated, while cord blood RvD1 and RvD2 levels were also positively correlated (Figure 1D).

### 3.3. Maternal and/or Cord Blood Plasma SPM Levels Are Associated with High-Risk Delivery and Maternal-Fetal Health Scenarios

We assessed whether RvD1 and RvD2 maternal or cord blood levels were associated with maternal-fetal health risks. As shown in Figure 2, when comparing maternal or cord blood plasma RvD1 and RvD2 levels with infant NICU admissions following delivery, maternal RvD1 (Figure 2A), maternal RvD2 (Figure 2B), and cord RvD2 (Figure 2D) levels were significantly elevated in samples from mother-infant pairs where infants were admitted to the NICU upon delivery. In pregnancies resulting in preterm delivery (defined in this study as delivery at less than 36 weeks infant gestational age), RvD1 and RvD2 maternal plasma levels were significantly elevated compared to women delivering at term (36 weeks or greater gestational age; Figure 3A,B). Cord blood RvD2 levels were also significantly increased in pairs delivering preterm (Figure 3D). To assess the utility of maternal RvD1 and RvD2 as biomarkers for preterm delivery and NICU admission, we generated Receiver-operating characteristic (ROC) curves. The area under the ROC curves for RvD1 and RvD2 levels in preterm vs. term deliveries were found to be 0.68 (*p* = 0.02) and 0.72 (*p* = 0.007), respectively (Figure 4A). The area under the ROC curves for RvD1 and RvD2 levels in NICU admission were 0.62 (*p* = 0.03) and 0.68 (*p* = 0.003), respectively (Figure 4B).

### 3.4. Association between Maternal n-3 Intake and SPM Levels in Maternal and Cord Blood

RvD1 and RvD2 are biosynthesized from the n-3 fatty acid DHA. When comparing SPM levels in mothers who responded to an FFQ including DHA intake data, we identified that in mothers whose infant was admitted to the NICU upon delivery, there was a positive association between increased DHA intake and maternal plasma RvD1 and RvD2 levels (Figure 5A,C), but not in mothers whose infants were not NICU-admitted (Figure 5B,D).

## 4. Discussion

In these investigations, we identified increased circulating levels of the omega-3 fatty acid derived SPM RvD1 and RvD2 in maternal peripheral blood plasma upon admission for delivery compared to infant cord plasma levels. In addition, significant associations were found between increased maternal plasma levels of RvD1 and RvD2 and cord plasma levels of RvD2 and high-risk maternal-fetal health conditions, including birth at less than 36 weeks estimated gestational age and infant NICU admission following delivery. We also identified that increased DHA intakes were associated with increased SPM levels in these at-risk scenarios. These findings suggest an important physiological role for SPM during human pregnancy and delivery, and also indicate a potential relationship between increased n-3 fatty acid intake, maternal SPM production, and high-risk maternal-fetal health conditions.

During inflammatory cascades, hydrolysis of n-3 or n-6 fatty acids from cell membranes and subsequent modifications (e.g., oxidation and/or elongation reactions) yield bioactive lipid metabolites that have pro-inflammatory and pro-resolution effects; temporal regulation of these mediators is critical in regulating inflammation resolution and wound repair [7,13,14,17]. Recently, SPM have been identified at high levels in human breast milk during the first month of lactation [13,14], and the resolvin SPM pathway precursor 17-HDHA (generated from the oxidation of DHA), was identified as being elevated in both maternal and cord blood plasma when measured at 34–36 weeks gestation and at delivery, respectively [21]. Our findings of high maternal RvD1 and RvD2 at delivery provide additional corroboration for the elevation of these lipids during gestation and delivery. Interestingly, we identified that maternal RvD1 and RvD2 levels were significantly higher than cord blood levels, which has not been previously reported. In addition to this finding, we have also identified associations between high maternal or cord blood SPM levels and adverse maternal-fetal outcomes. We found significant associations between high RvD1 and/or RvD2 levels and birth at less than 36 weeks gestational age and infant NICU admission upon birth. These findings suggest SPM may be elevated in these mothers or infants in response to inflammation or other adverse processes associated with these outcomes, which aligns with the known role of SPM in the active resolution of inflammatory environments [7,13,14]. It is important to note that many of the mother-infant pairs in our study who were NICU-admitted were also born preterm, thus these comparisons are recognizably overlapping and our findings should be considered bearing this in mind. Furthermore, our N for these comparisons are low, limiting the conclusions that can be drawn from these findings. Bearing these limitations in mind, our ROC curve findings generated from maternal RvD1 and RvD2 levels and preterm delivery or NICU admission do suggest that increased SPM levels could have utility as a biomarker for negative maternal-fetal outcomes.

The derivation of SPM primarily from n-3 fatty acids suggests a mechanism for benefit from n-3 supplementation during pregnancy. Indeed, previous reports have identified increased SPM precursors in the maternal blood, cord blood, and placental tissue in women with higher n-3 fatty acid intake or in those taking n-3 supplements [17,21,22]. Furthermore, in a preclinical study addressing the role of soybean oil-based lipid emulsion (SOLE) feeding versus fish oil-based lipid emulsion (FOLE) feeding, mice fed the FOLE exhibited higher circulating levels of SPM [16]. In the same study, five infants switched from the SOLE to the FOLE also experienced enhanced production of anti-inflammatory and pro-resolving lipid mediators in their serum [16]. It has been reported that DHA intake is associated with anti-inflammatory activities and with the prevention of numerous maternal-fetal health risks [1,2,3,20,23,24,25], and it has also been reported that SPM are produced during inflammatory responses [8,26]. Based on this, it is reasonable to postulate that sufficient intake of DHA may be required to create the repositories of SPM precursors, which can later be readily utilized during times of inflammation to produce SPM [21,22,27]. In support of this hypothesis, we identified a positive association between increasing maternal DHA intake and increased in RvD1 levels in maternal plasma in the setting of NICU admissions. This same trend was not seen in maternal-fetal pairs that were not admitted to the NICU upon delivery. It is notable that we identified this trend even though there was a potential deficiency of DHA intake in our population, whereby mean DHA intake was 119 mg/day, while the recommended minimum DHA intake is 300 mg/day for pregnant women, as suggested by the Workshop on the Essentiality of and Recommended Dietary Intakes for Omega-6 and Omega-3 Fatty Acids [20]. Using the National Health and Nutrition Examination Survey data, we and others have previously identified consistently low intakes of DHA in pregnant women and women of childbearing age in the US, with intake levels that closely reflect levels identified in our current study [18,19].

Our investigations utilized a commercially available enzyme immunoassay kit for determining RvD1 and RvD2 levels. Previous investigations have compared SPM levels taken from adult plasma (including EDTA collection, as in our investigation) using detection by liquid chromatography-tandem mass spectrometry whereby levels of RvD1 and RvD2 ranged from 2.6 pg/mL and undetectable in SRM 1950 NIST reference plasma (a composite of 100 healthy individuals) to 31.4 pg/mL and 26.4 pg/mL (respectively) in fish oil-supplemented individuals [15,28]. In addition, one published report has assessed serum SPM levels in five infants following being switched from a SOLE to FOLE diet [16]. One infant (born premature with respiratory distress syndrome and a history of necrotizing enterocolitis) exhibited a >1000-fold increase in serum RvD1 levels at 4 weeks following the lipid emulsion switch, with identified RvD1 levels around 1,400,000 pg/mL after one month of the fish oil-based lipid emulsion. This study substantiates the physiological relevance of the RvD1 and RvD2 levels we have identified, and also supports our finding of increased SPM levels in association with at-risk fetal conditions. We are unaware of published reports directly assessing plasma SPM levels measured using the enzyme immunoassay method of detection in a side-by-side comparison to measurement by liquid chromatography-tandem mass spectrometry. Although, one investigation has utilized the enzyme immunoassay method to measure RvD1 levels in serum of a limited number of healthy controls, and found samples to fall within a range of ~75–200 pg/mL [29]. This range falls below the RvD1 levels in maternal plasma found in our study, but does more closely correspond with our findings in cord blood plasma. This suggests that maternal plasma levels of RvD1 could be elevated, with cord blood levels falling within a normal range.

## 5. Conclusions

Taken together, our findings indicate an important biological role for the SPM RvD1 and RvD2 in labor and delivery and suggest a role for these mediators in responding to high-risk maternal-fetal health conditions. Future investigations are necessary to assess the utility of these SPM as biomarkers for these at-risk maternal-fetal health conditions, as well as to substantiate an n-3 fatty acid intake sufficiency threshold for optimal SPM production during neonatal inflammatory events. Taken together, futures studies warranted to understand how n-3 supplementation during pregnancy may alter SPM levels during labor/delivery, and/or protect against high-risk maternal-fetal outcomes.

## Figures and Tables

**Figure 1 nutrients-11-00098-f001:**
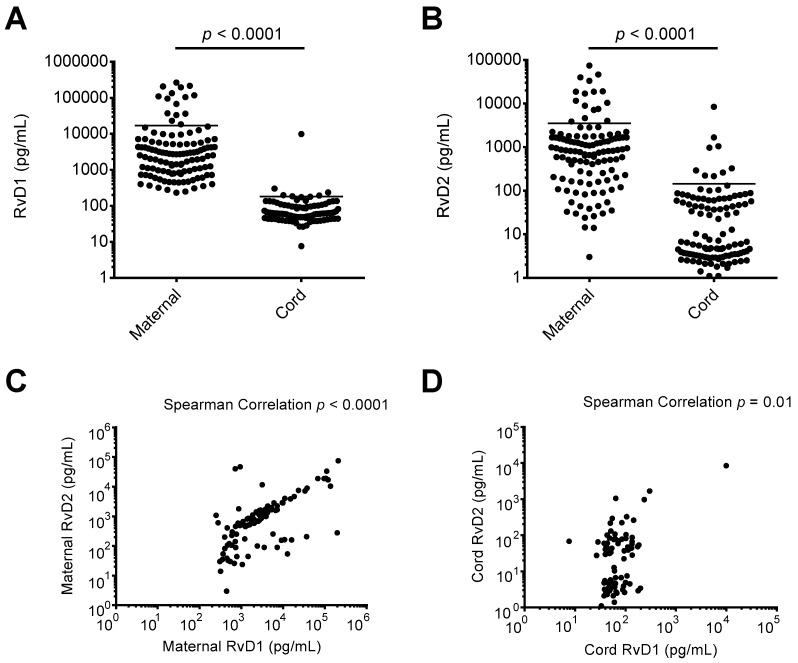
Resolvin D1 (RvD1) and Resolvin D2 (RvD2) levels in maternal and cord blood plasma. Enzyme immunoassays were performed on maternal and cord blood plasma samples collected upon admission for delivery and at birth, respectively. (**A**) RvD1 levels in maternal and cord blood plasma; (**B**) RvD2 levels in maternal and cord blood plasma; (**C**) comparison of maternal RvD1 and RvD2 plasma levels; (**D**) comparison of cord RvD1 and RvD2 plasma levels.

**Figure 2 nutrients-11-00098-f002:**
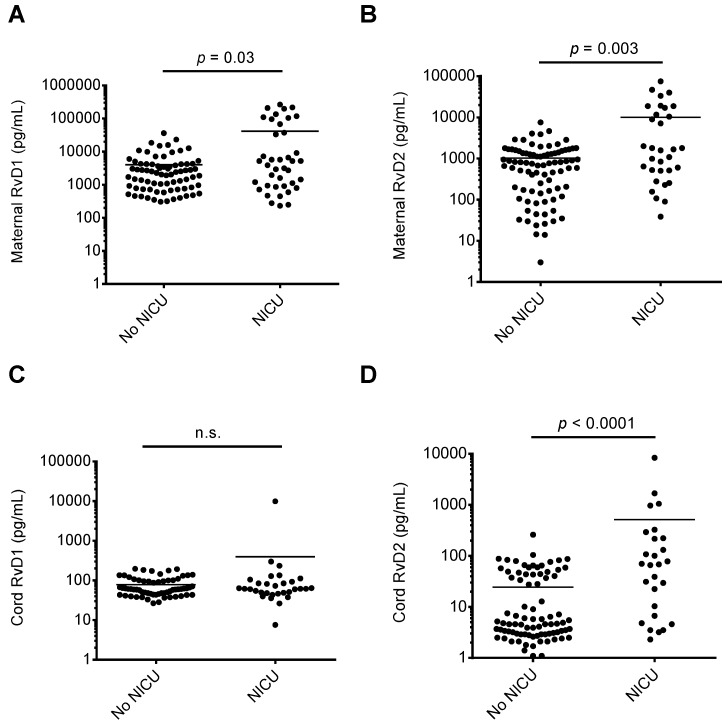
Comparison of maternal and cord blood plasma RvD1 and RvD2 levels and infant NICU admission at birth. Samples were stratified based on infant NICU admission at birth (yes/no) and maternal and cord blood plasma RvD1 and RvD2 levels were compared. (**A**) Maternal RvD1 levels; (**B**) maternal RvD2 levels; (**C**) cord RvD1 levels; (**D**) cord RvD2 levels. n.s.: no significant difference.

**Figure 3 nutrients-11-00098-f003:**
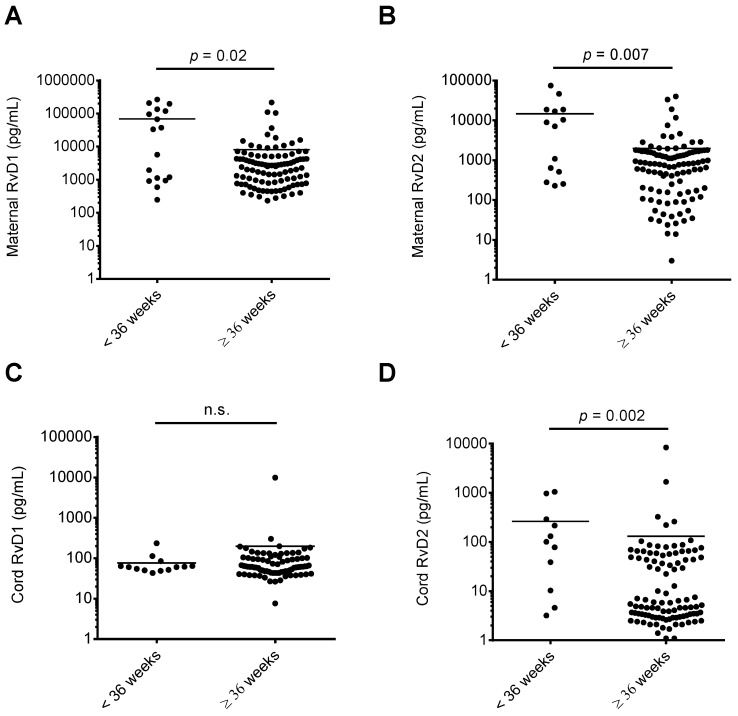
Comparison of maternal and cord blood plasma RvD1 and RvD2 levels and gestational age at delivery. Maternal and cord blood RvD1 and RvD2 levels were separated into infants delivering at less than 36 weeks estimated gestational age versus infants delivered at greater than or equal to 36 weeks estimated gestational age and compared. (**A**) Maternal RvD1 levels; (**B**) maternal RvD2 levels; (**C**) cord RvD1 levels; (**D**) cord RvD2 levels. n.s.: no significant difference.

**Figure 4 nutrients-11-00098-f004:**
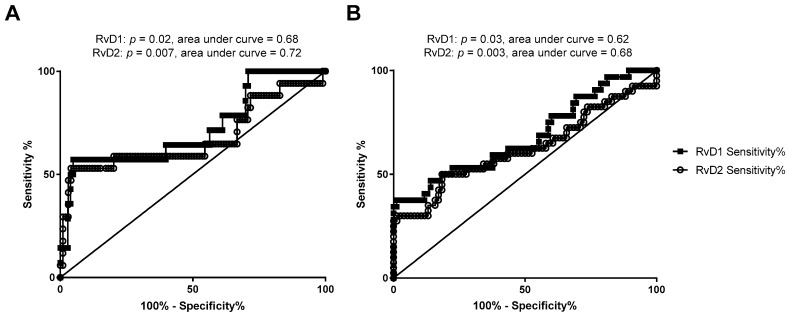
Receiver-operating characteristic (ROC) curves of sensitivity and specificity of maternal RvD1 and RvD2 in discriminating preterm birth, defined as infants delivered at less than 36 weeks estimated gestational age versus infants delivered at greater than or equal to 36 weeks gestational age (**A**), and NICU admission (**B**) outcomes.

**Figure 5 nutrients-11-00098-f005:**
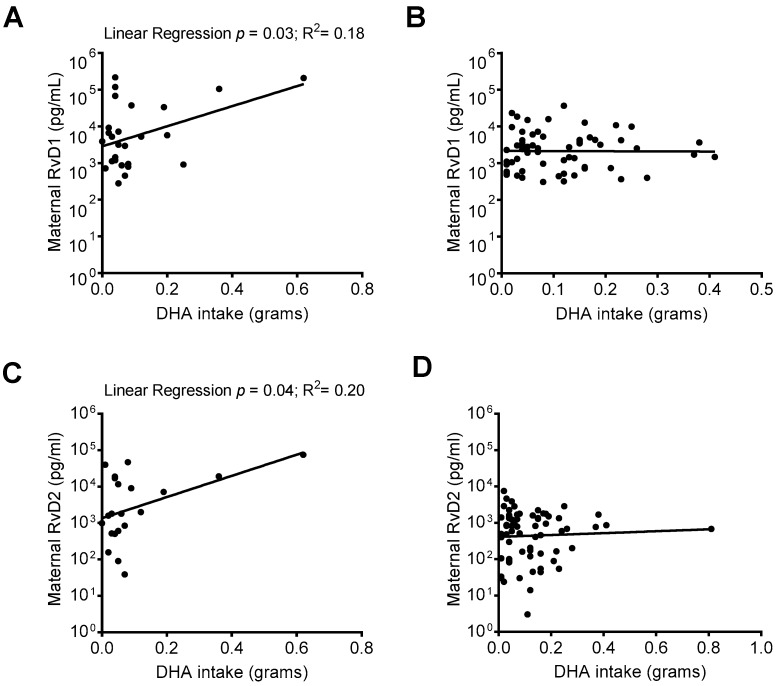
Relationship between DHA intake and maternal RvD1 and RvD2 plasma levels. Maternal RvD1 and RvD2 plasma levels were stratified based on infant NICU admission upon delivery and compared with DHA intake. (**A**) Linear regressions of maternal RvD1 levels and DHA intake in mothers with (**A**) and without (**B**) infant NICU admission upon delivery; Linear regression of maternal RvD2 levels and DHA intake in mothers with (**C**) and without (**D**) NICU admission upon delivery.

**Table 1 nutrients-11-00098-t001:** Participant Characteristics.

**Maternal/Infant Characteristics**	**N**	**Mean (SD)**
Maternal Age, years	136	28.9 (5.55)
Maternal pre-pregnancy BMI, m/kg^2^	80	26.9 (6.64)
Maternal DHA intake, grams	100	0.119 (0.131)
Birth EGA, weeks	138	38.2 (3.04)
Infant Birthweight, grams	138	3176.1 (704.8)
Infant Birth Head Circumference, cm	138	33.7 (2.79)
Infant Birth Length, cm	138	48.6 (4.84)
	**N**	**%**
Maternal n-3 Supplement Use (Y/N)	23/104	18/82
Infant Sex (M/F)	71/67	51/49
Infant NICU Admission at Birth (Y/N)	40/98	29/71
Placental Chorioamnionitis (Y/N)	20/118	14/86
Infant Antibiotic Use at Birth (Y/N)	26/112	19/81
Infant Respiratory Distress Syndrome (Y/N)	17/121	12/88

BMI: Body Mass Index; DHA: Docosahexaenoic Acid; SD: Standard Deviation; EGA: Estimated Gestational Age; Y/N: Yes/No; M/F: Male/Female.

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
