# Peer review of "Omega-3 Fatty Acid Supplementation, Pro-Resolving Mediators, and Clinical Outcomes in Maternal-Infant Pairs"

_nutrients, 2019, doi:10.3390/nu11010098_

Reviewer 1 Report

This is an interesting and informative study but there is information missing and some concern about the population enrolled.

1.       Information as to the patient recruitment is lacking.  Please provide the IRB#, the hospital at which the patients were recruited, the enrollment goals and how were they calculated (i.e. were power calculations performed to determine the size of enrollment), what geographic area was included.  It is also odd that so few of the mothers were receiving vitamins with DHA supplement as this is included in most prenatal vitamins.  Is there information on prenatal care?

2.       There was an unusually high percent of infants admitted to the NICU.  Is there an explanation for this? Were these for prematurity or other birth complications?

 3.       The correlations between RvD1 and RvD2 seem unnecessary and provides no new information as it would be expected that products of the same precursor molecules would be correlated with each other.

 4.       There is a contradiction if the RvD levels are higher in plasma from moms or infants admitted to the NICU but yet lower in infants ≥36 weeks.  This would indicate that NICU admissions might be for reasons other than prematurity and should be defined.  A correlation between GA and RvD levels would be informative.

 5.       Extensive discussion of the RvD precursors is distracting since these were not measured.  The discussion could be condensed by only mentioning the precursors as a potential mechanism, removing information already provided in the Introduction, and decreasing the discussion on the differences between the ELISA methods and the mass spec methods as those are accepted.

Author Response

1.       Information as to the patient recruitment is lacking.  Please provide the IRB#, the hospital at which the patients were recruited, the enrollment goals and how were they calculated (i.e. were power calculations performed to determine the size of enrollment), what geographic area was included.  It is also odd that so few of the mothers were receiving vitamins with DHA supplement as this is included in most prenatal vitamins.  Is there information on prenatal care?

    RESPONSE: Thank you for bringing up this concern. We have added the additional requested details in the methodology section of the manuscript. This project was conducted at the University of Nebraska Medical Center (UNMC) and its’ clinical partner, Nebraska Medicine. UNMC is the only public academic health science center in the State of Nebraska, and the major teaching and referral center serving rural and urban populations in Nebraska and the Midwest. The IRB for the University of Nebraska Medical Center provided human subjects approval under IRB# 115-12-EP. No power analysis for overall study recruitment was calculated, as this is an ongoing longitudinal cohort implemented with the intent of assessing and monitoring nutritional status in a Midwestern maternal-child cohort.

    With regard to the information on mothers receiving vitamins containing DHA, the information on prenatal vitamins was collected from the medical record, which includes prenatal visits. As a note, historically the use of prenatal vitamins in women of childbearing age in Nebraska is very low (32%), as described in the Nebraska Department of Health & Human Services 2012 Pregnancy Risk Assessment Monitoring System (PRAMS) Preconception Health Fact Sheet:

https://urldefense.proofpoint.com/v2/url?u=http-3A__dhhs.ne.gov_publichealth_WHI_Documents_PRAMS-5FFact.pdf&d=DwIGaQ&c=LsLxleeqPm1pgCNn-PN_bQ&r=XycsW29Y4TGoOKQaNrhzLCo49xiNayLYeY1luYPG1Ds&m=Na2q0GSXoKCaSCdLrj9bSg2MBEDO4jMdrp8_nZj4dvc&s=i39iOnfJgs-a7w86YSyybfa5ve-lw7tnCkQ4huW05Bw&e=.

2.       There was an unusually high percent of infants admitted to the NICU.  Is there an explanation for this? Were these for prematurity or other birth complications?

    RESPONSE: UNMC and Nebraska Medicine, as the only public academic health science center in the State, is the referral site for complicated pregnancies and high-risk deliveries. As a result, our NICU admission rates are higher than hospitals with a lower acuity delivery population. In addition, recruitment included oversampling of NICU admissions for sample size purposes. We have added this information into the methodology section of the manuscript.

3.       The correlations between RvD1 and RvD2 seem unnecessary and provides no new information as it would be expected that products of the same precursor molecules would be correlated with each other.

    RESPONSE: Thank you for raising this concern. We do agree that, as the reviewer has described, the RvD1/D2 levels would generally be expected to be correlated because of their formation from the same precursor fatty acid. For this reason, the correlation between these two mediators could be considered supportive confirmation of the validity of our assay results. However, there have been findings that the resolvin D-series mediators follow different temporal patterns during inflammation/resolution (e.g. Dalli, et al. Chem Biol 2014; PMID 23438748). Thus, we felt it was prudent to assess whether the RvD1/RvD2 were correlated within the context of this study.

4.       There is a contradiction if the RvD levels are higher in plasma from moms or infants admitted to the NICU but yet lower in infants ≥36 weeks.  This would indicate that NICU admissions might be for reasons other than prematurity and should be defined.  A correlation between GA and RvD levels would be informative.

    RESPONSE: We apologize for any lack of clarity in our description of the results based on gestational age at time of delivery. We have re-stated our findings in the Results section to clarify that preterm delivery was associated with higher maternal RvD1, maternal RvD2, and cord RvD2, which corroborates our findings that NICU admission (including preterm birth) was also associated with elevated levels of these same mediators.

5.       Extensive discussion of the RvD precursors is distracting since these were not measured.  The discussion could be condensed by only mentioning the precursors as a potential mechanism, removing information already provided in the Introduction, and decreasing the discussion on the differences between the ELISA methods and the mass spec methods as those are accepted.

    RESPONSE: Thank you for this suggestion, and we have revised the discussion of the manuscript to condense these sections.

Reviewer 2 Report

This study by Nordgren et al provides insight into the important role of SPMs in pregnancy and delivery risk.  The study has merit and could provide important knowledge, however a few concerns should be addressed prior to publication.

Major:

A few conceptual concerns are noted that tend to perpetuate misunderstandings regarding the role of lipid mediators in fluid and tissue compartments.  The authors are unclear about what they are measuring and appear to give little consideration to it.  Their methodology suggests they are measuring the unesterified compartment, but this is not explained, nor are any implications discussed.  For example, lines 206 discuss precursors to SPMs without any discussion of the likely tissue compartment they derive from.   

Second, it is not clear to this reviewer that SPMs include only n3-derived fatty acids.  This should be clarified and acknowledged.  Numerous n6 metabolites are involved in resolving inflammation and the authors should avoid conflating SPMs with omega-3 metabolites.

It is unclear why the authors are speculating about the utility of these metabolites as biomarkers when the data appears sufficient for an actual analysis.  Why not do it?  What is the c-stat? The reclassification rate? etc.

Comment on approach: the report would be greatly strengthened by including tissue FA enrichment data.

Other concerns:

Figure 1c & 1d appear to have multiple populations in the correlation plots.  These are likely explained by covariation.  Can the authors comments or provide insight?  Regardless, Spearman correlation would be more appropriate.

Line 112: are metabolites expressed?  I believe this confusing since it could be taken to invoke the canonical gene-mRNA-protein synthesis axis.

Figure 4: x-axes appear to lognormal.  I recommend transformation and/or Spearman correlation.

Author Response

Major:

A few conceptual concerns are noted that tend to perpetuate misunderstandings regarding the role of lipid mediators in fluid and tissue compartments.  The authors are unclear about what they are measuring and appear to give little consideration to it.  Their methodology suggests they are measuring the unesterified compartment, but this is not explained, nor are any implications discussed.  For example, lines 206 discuss precursors to SPMs without any discussion of the likely tissue compartment they derive from.   

    RESPONSE: Thank you for raising this concern. We have clarified in the manuscript to indicate that we are measuring oxylipins that are produced primarily through the oxygenation of unesterified DHA.

Second, it is not clear to this reviewer that SPMs include only n3-derived fatty acids.  This should be clarified and acknowledged.  Numerous n6 metabolites are involved in resolving inflammation and the authors should avoid conflating SPMs with omega-3 metabolites.

    RESPONSE: Thank you for this suggestion. We have modified the manuscript to indicate that certain SPM are derived from n-6 fatty acids.

 It is unclear why the authors are speculating about the utility of these metabolites as biomarkers when the data appears sufficient for an actual analysis.  Why not do it?  What is the c-stat? The reclassification rate? etc.

    RESPONSE: Thank you for this suggestion. We have now included additional statistical analyses to assess the utility of maternal RvD1/RvD2 as a biomarker for negative maternal-fetal health outcomes. Specifically, we have generated ROC curves (Receiver-operating characteristic curves) to assess the ability of the SPM to discriminate between term/preterm delivery or NICU admission/no NICU admission. We have updated the manuscript accordingly.

Comment on approach: the report would be greatly strengthened by including tissue FA enrichment data.

    RESPONSE: Thank you for providing this suggestion. While measurement for FA enrichment was not initially included in these studies, we recognized the importance of this outcome, and intend to include this measure in future studies.

Other concerns:

Figure 1c & 1d appear to have multiple populations in the correlation plots.  These are likely explained by covariation.  Can the authors comments or provide insight?  Regardless, Spearman correlation would be more appropriate.

    RESPONSE: Thank you for raising this question; it is possible that this finding of covariation is due to the different temporal patterns of resolvin-series mediators during inflammation/resolution (e.g. Dalli, et al. Chem Biol 2014; PMID 23438748), although this hypothesis remains to be explored. We have taken the reviewer’s suggestion and limited the statistics in Figure 1 to the Spearman correlation (removing linear regression analyses).

Line 112: are metabolites expressed?  I believe this confusing since it could be taken to invoke the canonical gene-mRNA-protein synthesis axis.

    RESPONSE: Thank you for pointing out this confusing language. We have removed the term “expression” in describing these mediators.

Figure 4: x-axes appear to lognormal.  I recommend transformation and/or Spearman correlation.

    RESPONSE: In the case of considering reported DHA intake levels (determined via food frequency questionnaires) as compared to our measured RvD1/RvD2 outcomes, we feel that the linear regression, whereby we are assessing the reliability of predicting our dependent variable (SPM) based on increasing levels of X (DHA), our independent variable, is appropriate. We log-transformed our dependent variable (SPM) to normalize the dataset for these analyses.

 Round  2

Reviewer 1 Report

The authors have responded appropriately to the reviewers suggestions.